# Algorithms for the selection of fluorescent reporters

Prashant Vaidyanathan [1,8], Evan Appleton [2,3,8], David Tran [4], Alexander Vahid[5], George Church [2,3] & Douglas Densmore [6,7 ✉]

Molecular biologists rely on the use of fluorescent probes to take measurements of their model systems. These fluorophores fall into various classes (e.g. fluorescent dyes, fluorescent proteins, etc.), but they all share some general properties (such as excitation and emission spectra, brightness) and require similar equipment for data acquisition. Selecting an ideal set of fluorophores for a particular measurement technology or vice versa is a multi-dimensional problem that is difficult to solve with ad hoc methods due to the enormous solution space of possible fluorophore panels. Choosing sub-optimal fluorophore panels can result in unreliable or erroneous measurements of biochemical properties in model systems. Here, we describe a set of algorithms, implemented in an open-source software tool, for solving these problems efficiently to arrive at fluorophore panels optimized for maximal signal and minimal bleed-through.

[1] Biological Computation group, Microsoft Research, Cambridge CB1 2FB, UK. [2] Department of Genetics, Harvard Medical School, Harvard University, Boston, MA 02115, USA. [3] Wyss Institute for Biologically Inspired Design, Harvard University, Boston, MA 02115, USA. [4] Alexa AI - Natural Understanding, Amazon, Cambridge, MA 02142, USA. [5] Amazon Global Logistics Tech, Amazon, Boston, MA 02210, USA. [6] Department of Electrical and Computer Engineering, Boston University, Boston, MA 02215, USA. [7] Biological Design Center, Boston University, Boston, MA 02215, USA. [8] These authors contributed equally: Prashant Vaidyanathan, Evan Appleton. ✉email: dougd@bu.edu

Fluorescence-based measurement of biological systems was first introduced in the 1960s and has since grown to be used as a method to measure biochemical properties in virtually every model organism[1]. These measurements have allowed researchers to probe the expression levels of proteins and other molecules at distinct points in time and space. Fields such as immunology, neuroscience, and synthetic biology make heavy use of these probes, as it is often critical to track expression of multiple signals within a sample. As such, fluorescence-based measurement has become the cornerstone of qualitative measurement technologies[2,3], quantitative measurement technologies[4] and modern sequencing technologies[5,6]. These technologies are built by composing expensive, precise mechanical equipment that relies upon lasers and detectors to measure fluorescence.

For all of these modern technologies, one key challenge is to maximize the number of different signals (i.e., colors) that can be distinguished in a single measurement in order to maximize the number of independent probes that can be used simultaneously, which can subsequently minimize the number of experiments and associated costs. However, choosing independent probes for a specific application or experiment is not trivial because many fluorescent probes emit light spectra that overlap with one another, making it difficult to separate signals from different probes (this problem is referred to as 'spectral spillover', 'bleed-through', or 'crossover'). To solve for potential bleed-through, makers of measurement machines often build in more lasers and detectors. While this gives the biologist more flexibility to choose probes, the onus is still on the user to resolve the trade-offs like bleed-through between certain probes. Fortunately, this problem of bleed-through can be corrected for by applying a series of linear algebraic operations by a process known as spectral compensation[7]. While the process of correcting spectral spillover using compensation is generally considered solved and easy to implement in theory, there are many nuances that make it difficult to implement in practice[8]. If compensation is misunderstood or misapplied, it can lead to incorrect biological measurements. As biologists perform experiments that need to be able to accurately resolve many fluorescent probes simultaneously[9–13] (where often >10 orthogonal probes are required), choosing the right set of fluorophores and corresponding detectors is crucial while designing experiments to ensure correct analysis of the resulting data[12].

In general, all fluorescence measurement technologies rely on the excitation of biochemical compounds with a laser (or other light source) at a certain wavelength and the capture of light emitted from these compounds across a spectrum of wavelengths using mirrors, filters, and light detectors. Current methods of designing large high-dimension flow cytometry panels involve painstaking ad hoc methods that involve computing spillover coefficients[14,15], analyzing gating strategies, and cycling through the latest literature to make qualitative-based selections. This is compounded by the fact that there isn't a 'one shoe fits all' solution for selecting n fluorophores out of a universal set of available fluorophores. This is primarily due to the fact that every measurement machine has a unique configuration (i.e., set of lasers and associated detectors). Exploring every possible panel design might be intractable depending upon the size of the panel. Hence, biologists and laboratories might be limited to panel designs that have already been studied and published or might have to limit the solution space based on an intuitive understanding of the emission and excitation spectra of the fluorophores available, to design a panel. The unfortunate consequence is that newly discovered fluorophores may not be easily incorporated in experiments since it may cause unintended spillover, forcing biologists to redesign panels from scratch.

To address this challenge, we created a computational solution housed in an open-source software tool to solve the following problem—given a measurement instrument where the laser and detector configurations are known, and a library of fluorophores under consideration, design an optimized n-color panel of fluorophores by maximizing the amount of signal in each detector and minimizing the amount of bleed-through. We also present a way to compare the optimality of two panels and present search algorithms to explore the solution space, find valid panel designs, and identify the best possible n-color fluorescence panel for a given instrument and collection of fluorophores. We demonstrate that our heuristic algorithm can reliably search for an optimal fluorescence panel even from a very large solution space. We also demonstrate that our computational predictions of signal and bleed-through reliably match experimental observations.

## Results

**FPselection.** To solve the fluorophore selection problem, we have designed an open-source web-application and command-line tool, to allow users to design an n-color panel for a specific measurement instrument - http://fpselection.org/. Our software uses search algorithms to explore the solution space of all possible n-panel configurations that can be constructed from a library of fluorophores for a fluorescence measurement instrument to find optimal panel configurations (Supplementary Section 1). The tool allows users to upload a file containing a list of fluorophores with excitation and emission spectra, an instrument configuration file, and choose a value for n to find valid and optimal n-color panels. Users can optionally upload files containing the brightness of each fluorophore or include the autofluorescence of the cell line used for the expression of the fluorescent proteins. Additional information regarding the format of these files can be found in Supplementary Section 2.1.

Since there are no established benchmarks for optimality of cytometry panels, we chose the following two properties to optimize an n-color panel: first, the amount of signal measured by a detector, for the fluorophore it is supposed to detect; and second, the amount of bleed-through from all other fluorophores in that detector. A panel configuration is considered as 'valid' if each detector in the panel measures a non-zero amount of signal for the fluorophore it is intended to measure (see Supplementary Section 1.1 for details on how predictions of signals and bleed-through are calculated for each detector in a panel). Finding a valid panel configuration is non-trivial as highlighted in Fig. 1, where the probability of randomly selecting a valid panel of 4 fluorophores out of a library of 8 for an instrument with 16 detectors is only 0.7%. An additional complexity is that not all valid panels can measure fluorescence efficiently. For example, Fig. 1 shows a valid panel containing a detector which has very low signal and high bleed-through from other fluorophores. One possible solution would be to use a larger library of fluorophores to get better valid panels that are optimal. However, the solution space increases exponentially with the increase in the size of the fluorophore library. For instance, the number of possible solutions to design a 4-color panel for the same measurement instrument used in Fig. 1 using all the available fluorophores listed in FPBase[16] (which has 734 fluorophores listed) is $5.23 \times 10^{14}$, which is clearly intractable. However, an algorithmic approach can find a reliable fluorophore panel from a reasonably large solution space. To solve for this problem, *FPselection* uses the search algorithms to find a valid panel where the amount of signal in each detector is maximized and bleed-through is minimized.

Figure 2 a shows that as the size of $n$ increases, and as the number of detectors in the cytometer increases, the percentage of

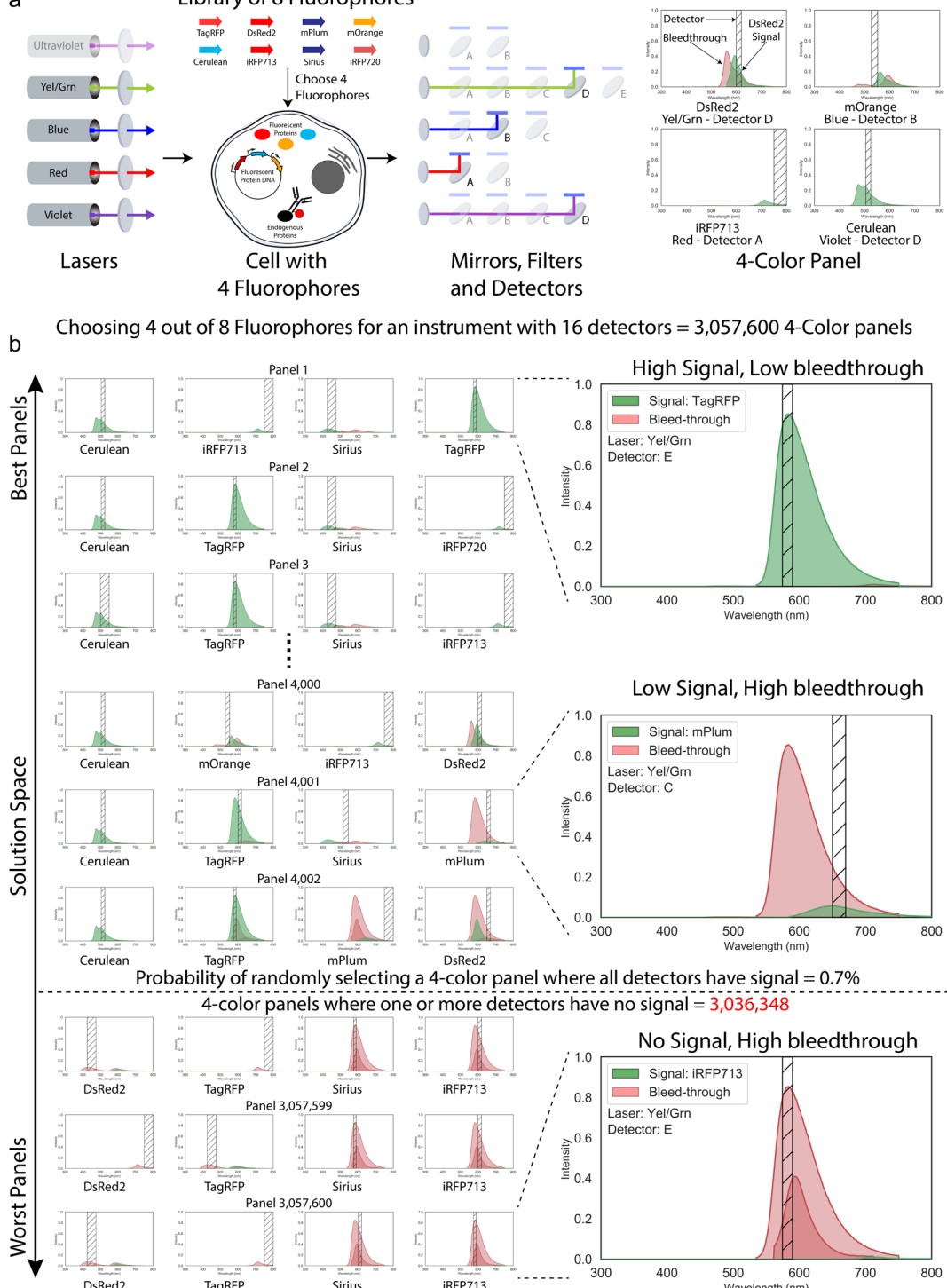

**Fig. 1 Selecting 4 fluorophores from a library of 8 fluorophores on a fluorometry machine with 16 detectors. a** Fluorophores can include fluorescent proteins, dyes, and conjugated antibodies. The light created by illuminating these fluorophores is generally activated with lasers of specific wavelength and captured by detectors that accept certain ranges of light signal on either a microscope or PMTs. The libraries of possible fluorophores to use and possible machine settings to detect the light are vast and present a problem difficult or impossible to properly optimize using ad hoc techniques. **b** In this case, selecting just 4 fluorophores from a library of 8 for this specific instrument is intractable since there are over 3 million possible 4-color panels, of which only 0.7% are valid. Even among the valid panels, a few configurations may be sub-optimal due to high bleed-through and low signal in the detector as shown in the middle plot on the right. Our algorithms can optimize the panel design by maximizing the signal in each detector and minimizing bleed-through (like the plot on the top right).

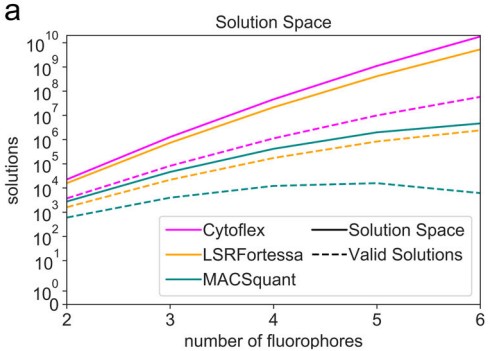
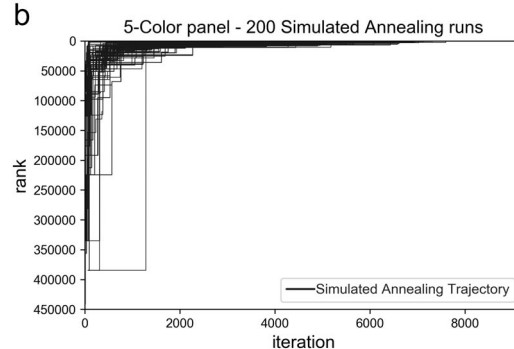

**Fig. 2 Computational performance of fluorophore selection algorithms. a** The number of possible solutions and valid solutions for selecting fluorophores from the library of 12 fluorophores (used in the case study) on any given machine increases exponentially with the size of the panel. This shows that for larger panel designs, it is not feasible to exhaustively search the entire solution space, and an efficient heuristic is required to find an optimal panel. **b** Trajectories of 200 runs of simulated annealing to search for a 5-color panel from a library of 8 fluorophores, for an instrument with 19 detectors. Each simulated annealing run has over 9000 iterations and the plot shows the rank of the best panel at any point of time during the run. The trajectories converge to the optimal solution (the best rank) toward the end of the run. This shows that the simulated annealing approach is a reliable heuristic to find an optimal panel design.

valid panel configurations also significantly reduces. For instance, the probability of randomly choosing a valid 2-color panel out of a library of 12 fluorophores for an instrument with 7 detectors is 21.75%. However, the probability of randomly choosing a valid 2-color panel from the same library for an instrument with 19 detectors is 10.17% and the probability of choosing a valid 6-color panel at random for the same instrument is 0.04%.

### Algorithm overview

*Searching for optimal n-color fluorophore panels.* We designed various search algorithms to explore the search space of n-color cytometry panels for a given set of fluorophores and instrument configurations. Supplementary section 1.4 contains the pseudo-code and performance metrics of these algorithms. Our recommended algorithm uses simulated annealing to quickly and reliably find an optimal result. The tool runs 50 concurrent threads, where each thread spawns a simulated annealing run, and the best result among the 50 threads is returned as the solution. This is particularly useful when the solution space is huge which can be due to larger fluorophore libraries, larger number of detectors in the measurement instrument, or larger values of n.

We tested the efficiency of our search algorithm by comparing it against an exhaustive list of valid panels for three different flow cytometers (BD LSRFortessa, Miltenyi MACSquant VYB, and CytoFlex LX) with varying and unique sets of lasers and detectors with the same sample sources to get experimental replicates. The detectors in all these cytometers were photomultiplier tubes (PMTs) with varying bandpass filters (Supplementary Section 3). For each cytometer, we performed 200 runs of simulated annealing to find n-color panels out of a library of 8 fluorophores (Supplementary Section 4), with the panel size ranging from 2 to 5. We then compared these results to an exhaustive list of all valid fluorophore panels ranked from best to worst, based on the criteria specified in the Methods section. The panel size and size of the library were restricted due to computational memory limitations, which were encountered while sorting the exhaustive list of valid panel designs.

We observed that the run-time for simulated annealing was constant and each run took <1 s to complete (as opposed to exhaustive search where the run-time increases exponentially with panel size and the size of the library and set of available detectors, and can take hours or days based on the size of the solution space—Supplementary Fig. 2). We also observed that simulated annealing typically returned the optimal solution in

most runs. For instance, Fig. 2b shows the trajectories of 200 runs of simulated annealing to search for 5-color panels, from a library of 8 fluorophores for a measurement instrument with 19 detectors. This test had the largest solution space, where there were 78,140,160 5-color panels of which 449,762 were valid. Simulated annealing was able to find a valid solution in all 200 runs and was able to find the best solution 191 times. The average rank of the solutions found by simulated annealing was 2.005 with a standard deviation of 15.718. This indicates that the simulated annealing approach performs reliably well as a heuristic and generally produces an optimal result.

**Comparing experimental observation and computational predictions.** To validate our computational predictions, we performed an experiment over-expressing our library of 12 fluorescent proteins in human induced pluripotent stem cells. We determined the signal and bleed-through values in each detector of the top 100 optimal n-color panels for all three cytometers for values of *n* ranging from 2 to 5. The signal and bleed-through (experimental) measurements and (computational) predictions in each detector were normalized to 1 based on the highest measured/predicted value respectively. In an ideal panel, the normalized value of the measured or predicted signal in each detector should be 1 while the normalized bleed-through values from other fluorophores should be as close to 0 as possible.

To compare the computational predictions against experimental measurements, we considered 2 basic metrics: 1. the number of panels where, for all detectors in the panel, the fluorophores with normalized values equaling 1 matched; and 2. the number of panels where for all detectors in the panel, the difference between each normalized prediction and measurement value must be within 0.05 (5%), 0.10 (10%), and 0.20 (20%) of one another. For instance, in Fig. 3a the signals match for all detectors, since for each detector, the fluorophore signal measurement and prediction values are 1. Hence, we say that the signals match for this panel. Similarly, the difference between normalized values of the signals and bleed-through are within 0.05 of each other (and hence by extension within 0.10 and 0.20 of each other) except the bleed-through from mPlum in the Yel/Green Laser's Detector C, where the normalized measurement and predictions have a difference of 0.123. Hence we say that this panel is within 0.20 but not within 0.05 or 0.10.

We started with a library of 12 fluorophores (see Supplementary Section 4 for the list of fluorophores used), where the

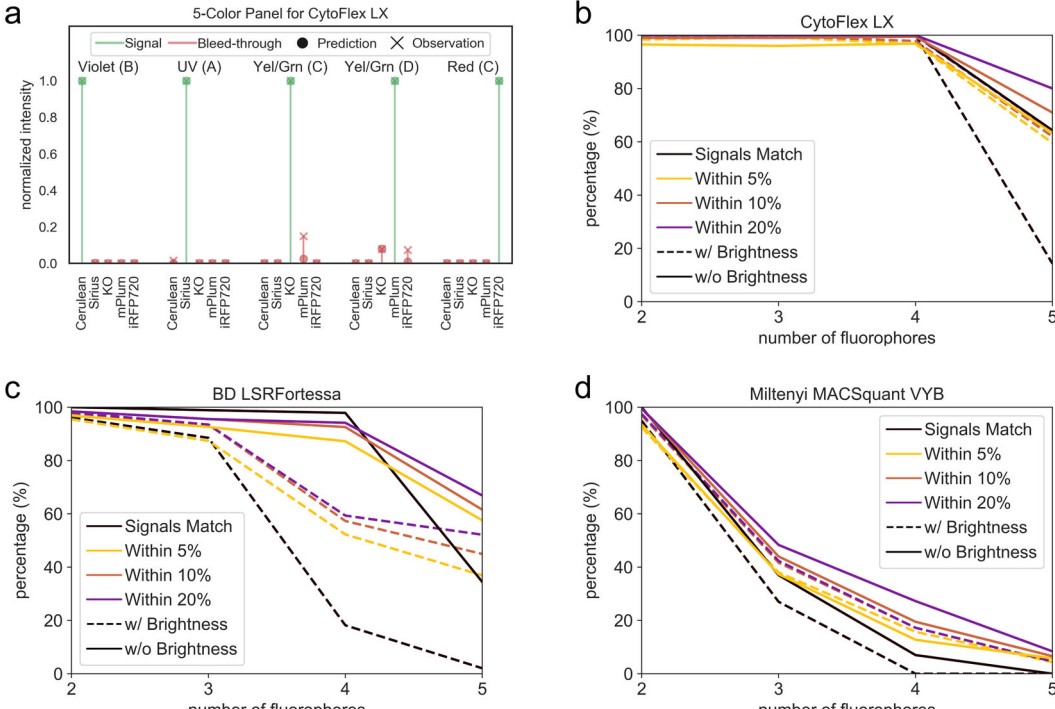

**Fig. 3 Experimental validation of exhaustive search fluorophore selection algorithm from a library of 12 fluorescent proteins expressed in human iPSCs on three flow cytometers. a** An example of the comparison between the computational predictions and experimental measurements for the best 5-color panel selected for the CytoFlex LX system, designed by fpSelection. The green lines indicate normalized signal values while the red lines indicate bleed-through from other fluorophores. In this panel, for each detector, the normalized values of signals and bleed-through match and are within 5% of one another except for the bleed-through from mPlum in detector C of the Yel/Grn laser where the difference between normalized computational and experimental intensity values is 12.3%. **b**, **c**, and **d** show the agreement of predicted signal/noise on a CytoFlex LX system with 5 lasers and 19 PMTs, on a BD LSRFortessa system with 5 lasers and 16 PMTs, and on a Miltenyi MACSquant system with 3 lasers and 7 PMTs respectively. The plots show the percentage of panels where the computational prediction of signal match the experimental measurements and the percentage of panels where the predicted signal and bleed-through values were within 5, 10, and 20% of the experimental measurements. The predictions are better when the emission of the fluorophores are not scaled based on their brightness. These plots also indicate that for larger panel sizes, having a larger variety of lasers and filters that can excite and measure wider range of wavelengths results in better agreement with computational predictions.

excitation and emission spectra and brightness were obtained from FPbase and other sources[16–19]. The term brightness, refers to the molecular brightness of a fluorophore which is the product of the molar extinction coefficient and the quantum yield of the fluorophore[16]. We normalized the brightness of the fluorophores to the brightest fluorophore - tdTomato (Supplementary Section 1.1). When we compared the predictions and measurements, we noticed that while predictions generally matched for 2 and 3 color panels, there was a significant drop for 4 and 5-color panels. We also noticed that the predictions matched more closely for the LSRFortessa and the CytoFlex (which had more lasers and detectors) as compared to the MACSquant which only had 3 lasers and 7 detectors.

Since the molecular brightness may not reflect the actual brightness observed in the experiments[16], we decided not to scale the emission of the fluorophores based on brightness for the next run. The predictions without brightness were significantly better than the previous run, especially for larger panel sizes. This is highlighted in Fig. 3b–d.

**Designing large n-color fluorophore panels**. One of the biggest motivations for developing heuristic algorithms to design multicolor fluorophore panels was to overcome the daunting challenge of designing large n-color fluorophore panels from a huge library of fluorophores. We used FPselection to design a 10-color fluorophore panel from a library of 188 fluorophores

(Supplementary Section 4) from FPbase, for the CytoFlex LX system (which has 19 detectors), without scaling the emission of the fluorophores based on brightness. The solution space of this particular problem is over $3.99 \times 10^{27}$ 10-color panels. Exploring such a large space exhaustively is computationally intractable. Even with access to parallel computing and better hardware, based on the estimates of running exhaustive search on design problems with smaller solution space, finding the optimal 10-color panel for an instrument with 19 detectors, out of a library of 188 fluorophores could take in the order of many millennia to solve.

We ran 5000 iterations of Simulated Annealing (with a higher starting temperature and lower cooling rate to account for the enormous solution space) and compared the best result from each iteration to find a solution. Figure 4 shows the computational prediction of the signal and bleed-through expected in each detector of the panel. While this solution is valid, it is computationally impossible to quantify how good this solution is compared to the optimal solution. However, by visually inspecting this panel, it is clear that 7 of the 10 detectors have relatively lower bleed-through, while the detectors assigned for miRFP720, mKelly2, and T-Sapphire will experience higher bleed-through from the other fluorescent probes. This observation was validated by computing the computational prediction of signal and bleed-through in each detector (Supplementary Section 1.3), where for 7 of the 10 detectors, the sum of bleed-through from all other fluorophores was <10% of the signal measured by the detector. This however shows that FPselection

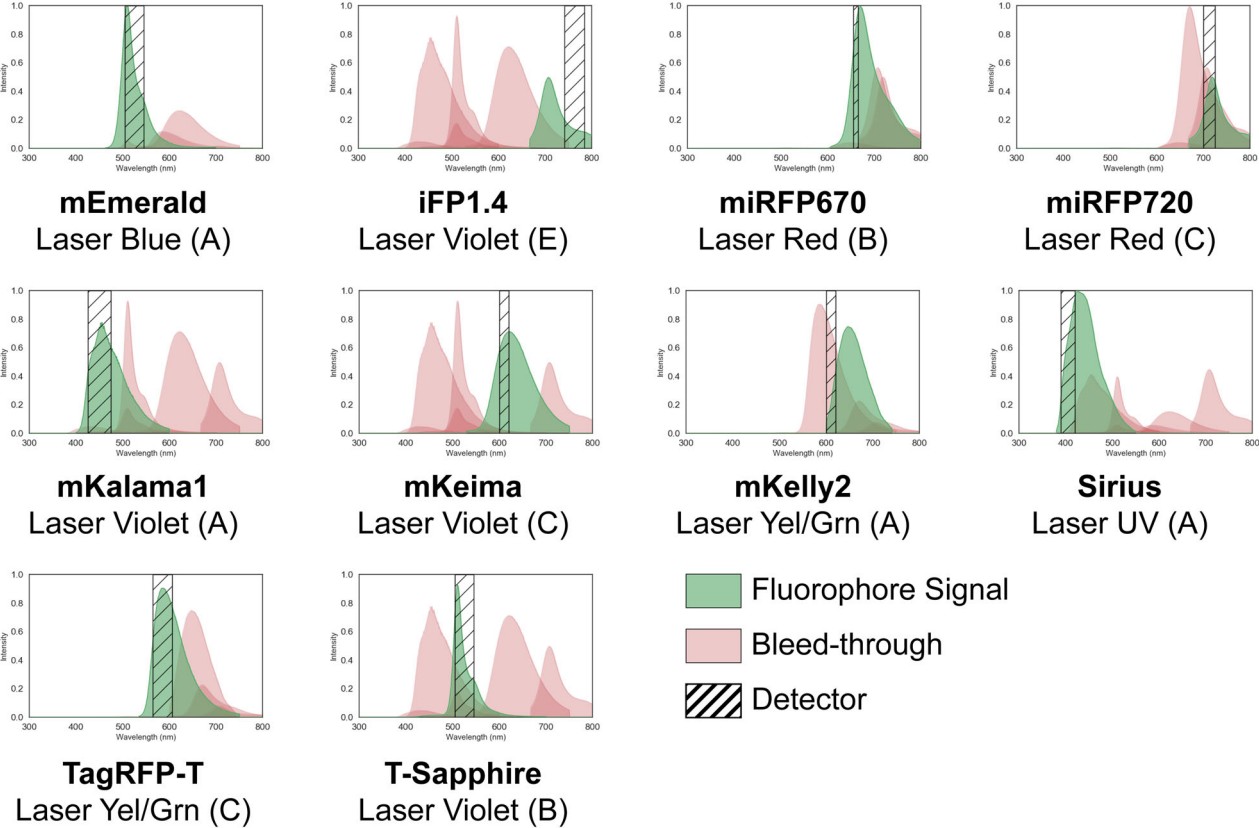

**Fig. 4 10-color fluorophore panel for the CytoFlex LX, selected from a library of 188 fluorophores obtained from FPbase, after running 5000 iterations of simulated annealing.** Out of the 5000 iterations of simulated annealing, this panel configuration had the best rank. Due to the enormous size of the solution space, it is computationally intractable to verify the optimality of this 10-color panel. However, this is a valid panel, since each detector has a non-zero amount of signal from the fluorophore it is intended to measure.

can very quickly identify a reasonably good and valid panel configuration for larger multicolor panels while choosing fluorophores from a huge library.

## Discussion

This work presents a set of algorithms that solve a key experimental design problem routinely re-solved by bench scientists—the selection of fluorescent reporters based upon the measurement device available in the lab. We have demonstrated that for small libraries of fluorophores on simple devices, we can do a reliable exhaustive search for an ideal set of fluorophores (Supplementary Fig. 4), and that for a large library of fluorophores on a complex device, we can use fast heuristics that still perform very well to select an optimal set of fluorophores (Fig. 2b) for an n-color panel. Furthermore, to verify that our algorithms were working properly and selecting reasonable solutions, we validated that fluorescent proteins expressed in human iPSCs had highly-matching results for overlap matrices compared to the algorithm-predicted outcomes (Fig. 3a). We also used our algorithms to design a 10-color panel from a huge library of fluorophores and showed that FPselection can quickly identify a valid panel design, where for 7 out of 10 detectors, the aggregate bleed-through was predicted to be <10% of the signal measured by the detector (Fig. 4).

The fluorophores used in this case study were fluorescent proteins. However, this work can very easily be applied to any type of fluorophore (such as fluorescent dyes and conjugated antibodies) as long as the emission and excitation spectra are available. Conjugated antibodies are often used to stain a broad spectrum of cellular surface and intracellular marker genes. While there is a short handful of stains often used with conjugated antibodies, the immunostaining field has advanced far enough to where if an expert user is willing to spend more money, they can use cutting edge stains with engineered spectra and still use this tool to optimize the selection of stains for use on a flow cytometer, microscope, or some other fluorescence machine.

While we did not go into depth in how additional optimizations of fluorophore selection can be made for other physical properties such as oligomerization, protein maturation, etc., the ability to weight these considerations exists in the algorithmic framework. Another way to do this in the current tools without any sophisticated optimizations already is simply to consider smaller subsets of fluorophores that are relevant to the particular experiments being performed. Future work could also be done with more sophisticated parameters of specific machines that have non-linear behavior in terms of laser strength, PMTs sensitivity, etc., but this work might have diminishing returns and become highly specific to each machine. However, since the software tool is open-source, the code can easily be modified to tailor the results to account for specific biochemical behaviors and instrument parameters.

In summary, we demonstrated that we can make fast, highly-optimized, and accurate decisions about which fluorophores to use for a given experiment on a given machine. This tool will serve a need that has existed in the bio-science community for a long time—countless bench scientists have had to solve this exact problem over and over again with varying degrees of proper

background knowledge, possibly leading to many unfortunate selections of fluorophores for experiments that hurt their reliability and reproducibility. While the validation in this study was done using flow cytometers for their quantitative readouts, these algorithms are just as relevant for any microscope or sequencing technology that uses lasers to read out fluorescent signal. We think that this new resource will save countless hours of work resolving this problem for a multitude of experiments and yield better choices of fluorophores for experiments.

## Methods

**Comparing two n-color panels**. We developed a method to compare two n-color panels, to assess the optimality of multicolor fluorescence panels. The panels are compared using the following three properties in the order specified below:

1. Number of detectors where bleed-through is within some threshold ($\eta$). We define a threshold $\eta$ as an acceptable percentage of bleed-through in a detector from other fluorophores that can be tolerated compared to the signal of the fluorophore it is supposed to detect. A detector is said to have bleed-through within $\eta$ if, the sum of all the bleed-through in that detector is within $\eta$% of the signal measured by that detector. The panel with the most number of detectors where bleed-through is within $\eta$ is considered better. If this value is the same for both panels, we use the next property. For our case studies, we chose the value of $\eta$ as 10% (the value of $\eta$ can be modified in the code).
2. Geometric mean of signals - We use geometric mean since this describes the "central tendency" of values in a data set better. By using geometric mean over arithmetic mean, we avoid choosing cytometry panels where a single signal value measured may be large but the other values are too small. The panel with the larger geometric mean of signals is considered better. If this value is the same for both the panels, we move to the last property.
3. Arithmetic mean of bleed-through observed in the filters. The total bleed-through in a detector is computed by adding the bleed-through from all other fluorophores of the panel. The mean bleed-through is computed by calculating the arithmetic mean of the total bleed-through in all the detectors of the panel. The panel with the smaller arithmetic mean of bleed-through is considered better. If both panels have the same value, we consider both panels equivalent.

**Cell line creation**. Human iPS cell lines were used for expression of fluorescent proteins. Fluorescent protein DNA sequences were cloned into a plasmid backbone that had an upstream doxycycline (DOX)-inducible pTRET promoter and a downstream puromycin expression cassette. Genomic integration sites flanked these components on the plasmid. These plasmids were nucleofected into iPS cells and then cells containing the fluorescent proteins cassettes were selected with puromycin (Sigma Aldrich). These cell lines were expanded in mTeSR (StemCell Technologies) and then induced with DOX for four continuous days. Cells were then digested with TrypLE (Gibco) and fixed in BD Cytofix buffer (BD Biosciences), washed in DPBS (Gibco), and stored at 4 °C for up to 1 week.

**Flow cytometry**. Fixed iPS cell lines were re-suspended in DPBS (Gibco) and run through three flow cytometers: (1) BD LSRFortessa [5 lasers | 16 channels]; (2) Miltenyi MACSquant VYB [3 lasers | 7 channels]; and (3) CytoFlex LX [5 lasers | 19 filter channels]. At least 10,000 events per sample were acquired within the dynamic range of the PMTs. Spectral overlap was calculated using the R Bioconductor package.

**Reporting summary**. Further information on research design is available in the Nature Research Reporting Summary linked to this article.

## Data availability

The Flow Cytometry Standard (fcs) data files containing measurements from the three cytometers in the case studies, that support the findings of this study are available in GitHub[https://github.com/CIDARLAB/fpSelection/tree/master/resources/bleedThroughAnalysis][20] with identifier https://doi.org/10.5281/zenodo.4202910. The normalized values of the signal and bleed-through for each detector of the 10-color panel solution shown in Fig. 4 can be found in Supplementary Data 1.csv. The configurations of the measurement instruments used in the case studies are available in Supplementary Data 2.zip with detailed information in Supplementary Sections 2.1 and 3. The emission and excitation spectra and the brightness of the fluorophores used in the case study are available in Supplementary Data 3.zip with detailed information in Supplementary Sections 2.1 and 4. Details of all inputs used in the case studies are available in GitHub [https://github.com/CIDARLAB/fpSelection/tree/v1.0/fpSelection/caseStudies]. Source data underlying the graphs and plots in figures is available in Supplementary Data 4.zip.

## Code availability

The open-source GitHub repository (https://github.com/CIDARLAB/fpSelection[20]) contains the code to run both the web-application as well as the command-line FPselection tool. The repository also contains the input files as well as scripts to reproduce the results discussed in this paper. Details of the Graphical User Interface and the software technology are available in Supplementary Sections 2.2 and 2.3 respectively. The FPselection tool is hosted at http://fpselection.org/.

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

## Acknowledgements

The authors would like to thank Luis Ortiz for his extensive feedback on the paper. The authors would also like to acknowledge Radhakrishna Sanka, Marilene Pavan, Calin Belta, and Nicholas DeLateur for their valuable feedback and insight. This work was supported by NSF grant #1522074 and #1446607 and the DARPA ELM Program under contract W911NF-17-2-0079.

## Author contributions

P.V., E.A., and D.D. conceived the project. P.V., D.T., A.V., E.A, and D.D. designed the software. E.A. designed and performed experiments. P.V. and E.A. analyzed the results. D.T. and A.V. contributed equally. D.D. and G.C. supervised this project. P.V., E.A., D.T., and A.V., were students of Boston University and part of the Biological Design Center at Boston University when they started working on this project. All authors wrote and reviewed the paper.

## Competing interests

The authors declare no competing interests.

## Additional information

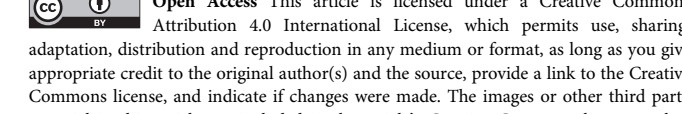

