## [Peer Review File · Communications Biology]

Reviewers' Comments:

Reviewer #1:

Remarks to the Author:

The manuscript, "Algorithms for the selection of fluorescent reporters", describes a software tool to search an optimized set of fluorescent probes from a panel of available fluorophores for biological samples measured by a flow cytometer. The searching is based on two properties (i.e. the amount of signal measured by a detector and the amount of bleed-through from all other fluorophores in that detector) using three algorithms, Exhaustive Search, Hill Climbing and Simulated Annealing in order to maximize the amount of signal in each detector and minimize the amount of bleed-through. This tool would be much more useful and significant for the field if authors can improve the software by adding a function mentioned below and provide additional results showing the searching results from Exhaustive Search are the same as that of Simulated Annealing, which is currently active online.

1. For biomedical researchers using a flow cytometer to measure multiple biomarkers (antigens) simultaneously with a set of fluorescent probes, the commercially available fluorescent probes conjugated with antibodies are limited. Most of time spectral bleed-through among probes is unavoidable. Since the software tool described here can already calculate bleed-through based on instrument setting among probes, it would be more helpful to users if the software can give a suggestion on compensation setup.
2. Figure 3 shows experimental validation of exhaustive search fluorophore selection algorithm. However the current active search algorithm online is Simulated Annealing. There is no data showing the searching results using different algorithms are the same. It is necessary to conduct more work to verify the consistency of searching results among different algorithms.

Reviewer #2:

Remarks to the Author:

Brief summary of the manuscript

The paper submitted by Vaidyanathan et al. presents an heuristic algorithm for the selection of fluorescent reporters in the context of single-cell analysis. For their study, the authors present a tool to enable biologists to design multi-colour fluorophore panels based on specific equipment's configurations. The authors demonstrate the efficacy of their algorithm by comparing computational predictions with experimental observations.

Overall impression of the work

The manuscript is clear, easy to read and concise. The purpose, methods and data analysis are clearly explained. I would recommend some changes.

Specific comments, with recommendations for addressing each comment

Major review:

- In the introduction it is said that "as biologists perform experiments that need to be able to accurately resolve many fluorophores probes simultaneously (when often greater than 10 orthogonal probes are required...", but on the other hand, paper shows examples of only 2 to 5-colour panels:

In page 3, comparing experimental observation and computational predictions, 2nd line: We determined the signal and bleed-through values in each detector of the top 100 optimal n-colour panels of all three cytometers for values of n ranging from 2 to 5.

Figure 1: Selecting 4 fluorophores from a library of 8 fluorophores

Figure 3D: An example of the comparison between the computational predictions and experimental measurements for the best 5-colour panel selected...

Figure S6: Result shown by fpselection.org for 2 fluorophore panel

It would be interesting to show, if possible, a bigger panel (10-colour maybe) as an example, as

researchers will be interested in using more probes in a single measurement.

- In the discussion section it is stated that “this work can easily be applied to any type of fluorophore (such as fluorescent dyes and conjugated antibodies)...”, however, only iPS cell lines that express fluorescent proteins are used for the paper. The information about the implications or limitations of using conjugated antibodies should be discussed in the discussion section. Things to consider: there is a limitation of colours for some markers, different antigen expression, some markers are co-expressed and some are non-coexpressed...

Minor points:

- In figure 1, I would recommend changing BOTTOM and TOP for A and B sections and modify the sections accordingly.
- Flow data has not been made available. Please provide links to data repositories (e.g. GitHub, FlowRepository, Cytobank, etc).
- Figure 1 should be mentioned in the text before Figure 2A.
- Figures 3A, 3B and 3C should be mentioned in the text before Figure 3D, or the figure should be arranged accordingly.
- Supplementary material should be organised in the order that it is referred in the main text (S1, S2, S3 and S4).

Fluorescent Protein Selection - Reviews

Prashant Vaidyanathan, Evan Appleton, David Tran, Alexander Vahid,
George Church, and Douglas Densmore

Author Comments Summary:

We'd like to thank the reviewers for their extremely valuable comments and feedback. We've incorporated all the feedback from the reviews in the manuscript as well as the software. To help navigate through the modifications and changes in the document, we have highlighted the changes in the main text and supplemental text in red font. We've also included page numbers and line numbers in both manuscripts to help identify the location of the changes.

Reviewer 1

The manuscript, "Algorithms for the selection of fluorescent reporters", describes a software tool to search an optimized set of fluorescent probes from a panel of available fluorophores for biological samples measured by a flow cytometer. The searching is based on two properties (i.e. the amount of signal measured by a detector and the amount of bleed-through from all other fluorophores in that detector) using three algorithms, Exhaustive Search, Hill Climbing and Simulated Annealing in order to maximize the amount of signal in each detector and minimize the amount of bleed-through. This tool would be much more useful and significant for the field if authors can improve the software by adding a function mentioned below and provide additional results showing the searching results from Exhaustive Search are the same as that of Simulated Annealing, which is currently active online.

1. For biomedical researchers using a flow cytometer to measure multiple biomarkers (antigens) simultaneously with a set of fluorescent probes, the commercially available fluorescent probes conjugated with antibodies are limited. Most of time spectral bleed-through among probes is unavoidable. Since the software tool described here can already calculate bleed-through based on instrument setting among probes, it would be more helpful to users if the software can give a suggestion on compensation setup.

Author Comments: We agree that suggesting compensation setup would be a useful feature for users. In our case studies, we used R scripts to help with compensation setup, and used these scripts to verify the accuracy of our computational predictions against experimental observations. Hence, we have added these scripts to the GitHub repository along with detailed instructions to use the scripts. These can be found via this link. We have also added a section in the tool to link to these scripts.

2. Figure 3 shows experimental validation of exhaustive search fluorophore selection algorithm. However the current active search algorithm online is Simulated Annealing. There is no data showing the searching results using different algorithms are the same. It is necessary to conduct more work to verify the consistency of searching results among different algorithms.

Author Comments: To address this, we created additional scripts that can help verify the consistency of the searching results of the heuristic algorithms. We ran 200 seeded runs each of Simulated Annealing and Hill Climbing. For each run, we spawned 50 seeded iterations of the respective algorithm. We compared the best results from the 50 iterations to get the best result of the run. This process, mirrors a normal execution of 1 run of the heuristic algorithm via the FPselection website (which is set to Simulated Annealing by default). We then compared the result of each run and verified that the the result was always within the top 100 results produced by exhaustive search. We ran these for all 3 cytometers, for panel sizes ranging from 2 to 5, and for both fluorophore libraries (of size 8 and 12). In every run, the result was always within the top 100. These files (as well as the script to recreate the results) have been uploaded to the GitHub repository. We have also added an additional section in the supplementary materials Page 9 - line 153 to line 164 to address this comment.

Reviewer 2

Brief summary of the manuscript The paper submitted by Vaidyanathan et al. presents an heuristic algorithm for the selection of fluorescent reporters in the context of single-cell analysis. For their study, the authors present a tool to enable biologists to design multi-colour fluorophore panels based on specific equipment's configurations. The authors demonstrate the efficacy of their algorithm by comparing computational predictions with experimental observations.

Overall impression of the work The manuscript is clear, easy to read and concise. The purpose, methods and data analysis are clearly explained. I would recommend some changes.

Specific comments, with recommendations for addressing each comment

Major review:

- In the introduction it is said that “as biologists perform experiments that need to be able to accurately resolve many fluorophores probes simultaneously (when often greater than 10 orthogonal probes are required...”, but on the other hand, paper shows examples of only 2 to 5-colour panels: In page 3, comparing experimental observation and computational predictions, 2nd line: We determined the signal and bleed-through values in each detector of the top 100 optimal n-colour panels of all three cytometers for values of n ranging from 2 to 5. Figure 1: Selecting 4 fluorophores from a library of 8 fluorophores Figure 3D: An example of the comparison between the computational predictions and experimental measurements for the best 5-colour panel selected... Figure S6: Result shown by fpselection.org for 2 fluorophore panel It would be interesting to show, if possible, a bigger panel (10-colour maybe) as an example, as researchers will be interested in using more probes in a single measurement.

Author Comments: We agree that this suggestion would be a great way to highlight the utility of the FPselection tool. Since designing 10-color panels typically involves an enormous solution space, we thought that this suggestion presented a unique opportunity for us to also demonstrate how FPselection can reliably design n-color panels from a huge library of fluorophores. To address this comment, we have added a new sub-section **Designing large n-color fluorophore panels** in the main-text (Page 4 of the main text - line 141 to line 159). In this case study, we created a library of 188 fluorophores from FPbase (details of this library can be found in the SI), and designed a 10-color panel for the CytoFlex LX system (which has 5 lasers and 19 detectors). We obtained a result by running 5,000 iterations of simulated annealing and comparing the best results in each iteration to get the best panel design. We have also added an image (Page 10 of the main text) showing the computational predictions of signal and bleed-through for each detector of the best 10-color panel designed in the case study as shown below.

Due to the enormous solution space (3.99×10^{27} solutions), it is not possible to run exhaustive search to validate the optimality/rank of this result. It is also worth noting that the solution space to design a 10-color panel from the original library of 12 fluorophores and the Fortessa cytometer (which has 16 detectors) is close to 2 trillion and hence is still intractable via exhaustive search. Hence, we decided to use a larger library to present a stronger case. We hope that this case study tackles two important questions:

- Can FPselection be used to design large n-color panels (in this case $n = 10$)
- Can FPselection reliably design fluorophore panels if the available library is huge (in this case 188 fluorophores).

We have also added an additional line in the discussion section to highlight this (Page 4 of the main text - line 167 to line 169)

- In the discussion section it is stated that “this work can easily be applied to any type of fluorophore (such as fluorescent dyes and conjugated antibodies)...”, however, only iPS cell lines that express fluorescent proteins are used for the paper. The information about the implications or limitations of using conjugated antibodies should be discussed in the discussion section. Things to consider: there is a limitation of colours for some markers, different antigen expression, some markers are co-expressed and some are non-coexpressed.

Author Comments: We have added additional lines in the Discussion section to address this. The additions can be found in Page 4 of the main text - line 172 to line 175.

Figure 1: 10-color fluorophore panel for the CytoFlex LX, selected from a library of 188 fluorophores obtained from FPbase, after running 5000 iterations of simulated annealing. Out of the 5000 iterations of simulated annealing, this panel configuration had the best rank. This is a valid panel, since each detector has a non-zero amount of signal from the fluorophore it is intended to measure. Out of the 10 detectors in the panel, the aggregate bleed-through in 7 detectors is less than 10% of the total signal measured by the assigned detector.

Minor points:

- In figure 1, I would recommend changing BOTTOM and TOP for A and B sections and modify the sections accordingly.

Author Comments: We’ve updated Figure 1 to now have sections A and B. We have also updated the figure caption accordingly. The updated figure can be found in Page 7 of the main text.

- Flow data has not been made available. Please provide links to data repositories (e.g. GitHub, FlowRepository, Cytobank, etc).

Author Comments: We have addressed this by uploading all flow data to the GitHub repository. More specifically, the data is now accessible the following location <https://github.com/CIDARLAB/fpSelection/tree/master/resources/bleedThroughAnalysis>. We have also added additional text in the Supplemental Information Document that provides links to the flow data (Supplementary material Page 14 Line 245).

- Figure 1 should be mentioned in the text before Figure 2A.

Author Comments: We have addressed this in the Main text - Page 2, line 45.

- Figures 3A, 3B and 3C should be mentioned in the text before Figure 3D, or the figure should be arranged accordingly.

Author Comments: We addressed this by rearranging the figures and the associated figure text. This can be verified in Page 3, lines 125 and 140 of the main text.

- Supplementary material should be organised in the order that it is referred in the main text (S1, S2, S3 and S4).

Author Comments: This has been addressed by rearranging the sections of the supplementary material, and by adding appropriate references to the sections in the main document.

REVIEWERS' COMMENTS:

Reviewer #1 (Remarks to the Author):

The revised version is satisfactory.

Reviewer #2 (Remarks to the Author):

Thanks for considering my suggestions. I am satisfied with the changes, so no more comments from my part.